# First Insights into Body Localization of an Osmoregulation-Related Cotransporter in Estuarine Annelids

**DOI:** 10.3390/biology13040235

**Published:** 2024-03-31

**Authors:** Serena Mucciolo, Andrea Desiderato, Maria Mastrodonato, Paulo Lana, Carolina Arruda Freire, Viviane Prodocimo

**Affiliations:** 1Department of Invertebrate Zoology and Hydrobiology, University of Lodz, Banacha 12/16, 90-237 Lodz, Poland; 2Laboratório de Bentos, Centro de Estudos do Mar, Universidade Federal do Paraná, Av. Beira Mar s/n, Pontal do Paraná 83255-976, Paraná, Brazil; lana@ufpr.br; 3Dipartimento di Bioscienze Biotecnologie e Ambiente, Campus Universitario “E. Quagliariello”, Università degli Studi di Bari Aldo Moro, via Orabona, 4, 70125 Bari, Italy; maria.mastrodonato@uniba.it; 4Laboratório de Fisiologia Comparativa de Osmorregulação, Departamento de Fisiologia, Setor de Ciências Biológicas, Campus Politécnico, Universidade Federal do Paraná, Av. Cel. Francisco H. dos Santos 100, Curitiba 81530-000, Paraná, Brazil; cafreire@ufpr.br (C.A.F.); vprodocimo@ufpr.br (V.P.)

**Keywords:** invertebrate, osmoconformer organisms, physiology, annelid, NKCC cotransporter

## Abstract

**Simple Summary:**

The study aimed to confirm the presence of the Na^+^-K^+^-2Cl^−^ cotransporter (NKCC) in annelids, a membrane transporter crucial for cell volume regulation, which had not been directly shown before in these worms. Initially, the presence of NKCC was first recovered in silico. Using immunofluorescence, we observed that the NKCC signal was detected in various tissues from four estuarine annelid species from southern Brazil subjected to salinity changes. Two euryhaline species and two stenohaline species were tested. While all species showed NKCC expression, its distribution varied. Euryhaline and free-living species showed widespread NKCC expression along their bodies, while the sedentary stenohaline species had it mainly localized in branchiae and internal tissues. These findings suggest NKCC plays a role in cell volume regulation, particularly in annelids experiencing habitat salinity fluctuations.

**Abstract:**

The expression of the Na^+^-K^+^-2Cl^−^ cotransporter (NKCC), widely associated with cell volume regulation, has never been directly demonstrated in annelids. Its putative presence was firstly recovered in silico, and then using immunofluorescence, its signal was retrieved for the first time in different tissues of four species of estuarine annelids from southern Brazil that are regularly subjected to salinity fluctuations. We tested two euryhaline species (wide salinity tolerance), the nereidids *Alitta yarae* and *Laeonereis acuta* (habitat salinity: ~10–28 psu), and two stenohaline species (restricted salinity tolerance), the nephtyid *Nephtys fluviatilis* (habitat salinity: ~6–10 psu), and the melinnid *Isolda pulchella* (habitat salinity: ~28–35 psu). All four species showed specific immunofluorescent labelling for NKCC-like expression. However, the expression of an NKCC-like protein was not homogeneous among them. The free-living/burrowers (both euryhaline nereidids and the stenohaline nephtyid) displayed a widespread signal for an NKCC-like protein along their bodies, in contrast to the stenohaline sedentary melinnid, in which the signal was restricted to the branchiae and the internal tissues of the body. The results are compatible with NKCC involvement in cell volume, especially in annelids that face wide variations in salinity in their habitats.

## 1. Introduction

Annelids occupy a wide range of environments (e.g., seawater, freshwater, soil), thanks to their high body plasticity and life strategies [1]. One of the biggest requirements for survival in different habitats is the capability to regulate the bodily internal osmotic pressure in the fluids. For instance, marine annelids are generally in osmotic equilibrium with saltwater, adjusting their internal osmotic pressure with the surrounding environment. Differently, freshwater and terrestrial animals usually maintain a moderate/high internal osmotic pressure and need to face water gain and desiccation stress, respectively [2]. Phylogenetic hypotheses, and the derived classifications of annelids, are constantly changing, due to both their diversification and adopted approaches. According to one of the most accepted arrangements (Figure 1A), Annelida are divided into two major clades, Sedentaria and Errantia, and five basal sister branches: Sipuncula, Amphinomidae, Chaetopteridae, Magelonidae, and Oweniidae [3]. Regarding the osmoregulation, no correlation appears between annelid classification and osmoregulatory strategies, displaying either osmoconformity or osmoregulation along the whole phylum. Sedentaria, for instance, includes mostly osmoconformer annelids, with the exception of the Clitellata, animals performing mostly osmoregulation and occurring in freshwater and terrestrial environments. On the contrary, Errantia, as well as the remaining clades, group together mostly marine osmoconformer annelids with sporadic examples of species capable of establishing osmotic gradients with the external environment e.g., [1,2,4,5]. The diverse osmoregulatory strategies (i.e., conformation or regulation) may be reflected also in the ecological and morphological differences that Sedentaria and Errantia present (Figure 1B,C). The tubicolous lifestyle of Sedentaria is reflected in reduced parapodia, which, on the contrary, are mostly well developed for their intense locomotory activity in Errantia (Figure 1B,C). Nonetheless, in addition to their main role in locomotion, parapodia are also involved in ion and gas exchanges [2]. Different organizations of the excretory system, such as the type and localization of nephridia (along the body in Errantia, while restricted to the first few anterior segments in tubicolous Sedentaria), may influence higher osmotic tolerance and osmoregulatory capability (Figure 1B,C) [6,7].

At the cellular level of body organization, different membrane proteins work together in order to regulate the cell volume, performing ion and water exchanges [12]. The Na^+^-K^+^-2Cl⁻ (NKCC), an electroneutral cotransporter, belongs to the family of cation-chloride cotransporters (CCC) and transports sodium, potassium, and chloride ions across the cell membrane. Two isoforms are known in literature, NKCC1 and NKCC2, the former occurring in the basolateral membrane of secretory epithelia and the latter only in the apical absorptive kidney epithelia [12,13]. NKCC1 plays a crucial role in vectorial salt excretion in vertebrates, as in the gills of marine teleosts, in the rectal gland of elasmobranchs, or in the salt gland of marine birds and reptiles [12,14,15]. The absorptive NKCC2, in turn, occurs in the apical membrane of the thick ascending limb of the loop of Henle in the mammalian kidney, where this cotransporter was discovered [16,17]. In addition to its occurrence in epithelia, NKCC1 is also widely reported in cells in general, acting effectively in volume control [12,13]. In vertebrates, it has been specifically related to regulatory volume increase (RVI) activated by cell shrinkage, allowing for significant solute influx, which is followed by water influx and cell volume restoration [12,18]. Conversely, NKCC1 seems to be activated in case of cell swelling, performing the efflux of osmolytes during Regulatory Volume Decrease (RVD), due to distinct ionic electrochemical gradients expected in osmoconforming invertebrates, such as molluscs and echinoderms [19,20,21]. As a transporter that relies on electrochemical gradients, it may operate reversibly, depending on the prevailing driving forces. In fact, depending on the anisosmotic or isosmotic conditions, it may operate in RVI or RVD, respectively, even in mammalian cells [12].

Estuarine annelids are frequently euryhaline, thus tolerating wide and fast salinity changes in their habitats, meaning that their cells are intensely challenged with respect to their volume [2,22,23]. Studies on the presence and functions of NKCC are scarce and only hypothesized in annelids. For example, Dykens and Mangum, 1984 [24] pointed out how the presence of Na^+^/K^+^-ATPase, occurring only in the superficially vascularized part of the parapodia, was not sufficient to explain the cell volume regulation widespread in the whole body. The potential presence and functionality of NKCC in osmoregulation/cell volume regulation of annelids was proposed for the leech *Hirudo medicinalis* Linnaeus 1758, based on the effects of its inhibitor furosemide in primary urine formation [14,25]. Nevertheless, no study has clearly demonstrated the presence of this membrane protein in the epithelia of annelids. Therefore, the aim of this work was to reveal the expression and localization of the NKCC cotransporter through immunofluorescence in the bodies of marine annelids that live under variable salinity challenges in estuarine habitats.

## 2. Materials and Methods

### 2.1. In Silico Search

To check for the presence of the putative NKCC cotransporter in the annelids *in silico*, a preliminary search of an internet database and a phylogenetic reconstruction was conducted. An amino-acid sequence of the NKCC-like protein per phylum was firstly selected from the comprehensive phylogenetic analysis on the entire CCC family carried out by Hartmann et al. 2013 [26]. Multiple blastp [27] were performed in the protein database of NCBI (accessed 10 October 2020) using each selected sequence as a query against the Annelida database (taxid 6340). For each protein, only sequences with an expectation value (e-value) of 0.0 were downloaded. All accession numbers are listed in the Appendix A. One sequence per species was aligned using MUSCLE [28] as implemented in MEGA7 [29]. A maximum likelihood (ML) tree was built using PhyML 3.0 [30], and the robustness of the nodes was estimated by 1000 bootstrap replicates. The best substitution model (LG + I + G) was tested with the SMS routine in PhyML 3.0, using Akaike Information Criterion (AIC) as optimality criteria [31]. Sequences of the choanoflagellate *Monosiga brevicollis* potassium-chloride cotransporter (KCC) and chloride transporter-interacting protein (CIP1) were used as the root of the phylogeny.

### 2.2. Animals

Sampling was carried out during the dry season (May–September 2018) along the Paranaguá Estuarine Complex (PEC) in southern Brazil, one of the most preserved coastal areas along the southwestern Atlantic, despite increasing port and tourist activities (Figure 2). The estuary is considered mixed, with seasonal pattern variations of salinity and temperature and an increasing salinity gradient going from the inner to the outer sectors [32]. Species were chosen considering their occurrence along the estuary and expected salinity tolerance: (a) the nereidids *Alitta yarae*, usually associated with hard and human-made substrates, and *Laeonereis acuta* (Treadwell 1923), in soft bottoms, both abundant in meso/polyhaline sectors of the estuaries habitat salinity range: ~10–28 psu), and putatively euryhaline; (b) the nephtyid *Nephtys fluviatilis* Monro 1937, from the oligohaline sectors of the estuary (habitat salinity range: ~6–10 psu), and the melinnid *Isolda pulchella* Müller 1858, usually related to poly/euhaline sectors of the bays (habitat salinity range: ~28–35 psu), both putatively stenohaline (Figure 2). Animals were sampled by using a shovel or scraping buoys, depending on the species. Authorization for animal sampling was provided by “Sistema de Autorização e Informação em Biodiversidade” (SISBIO, permit # 36255-1/73627-1).

### 2.3. Immunofluorescence

In the laboratory, the animals were acclimatized for 48 h in plastic containers of 1 L with water, and 3–4 cm of sediment collected from each sample area, under constant temperature (~20 °C), aeration and natural photoperiod, and fed with common aquarium flakes. Healthy adults were chosen as described in Mucciolo et al. 2021 [5]. Individuals of *A. yarae* and *L. acuta* were fixed in 2% paraformaldehyde diluted in 1% PBS for 2 h, while *I. pulchella* and *N. fluviatilis* were fixed in 2% paraformaldehyde in 1% PBS for 1.5 h because of their smaller size (~4–5 cm the nereidids, and ~1–2 cm, the melinnid and nephtyid). Animals were then washed in 1% PBS, incubated in 1% PBS with 5% sucrose for 2 h and finally in 1% PBS with 15% sucrose overnight at 4 °C. Specimens were embedded in Fisher Healthcare™ Tissue Plus™ O.C.T. compound and stored at −20 °C. Blocks were sagittally sectioned in the cryostat Leica CM1850, Heidelberger, Germany at −25 °C, with sections of 15–20 µm thickness for the species *A. yarae* and *L. acuta* and 5–8 µm for the species *I. pulchella* and *N. fluviatilis*, and placed on glass slides previously prepared with a solution of 2% 3-aminopropyltriethoxysilane diluted in acetone for section adhesion. Section washing and incubation followed the protocol from Prodocimo and Freire, 2006 [33] using the primary antibody anti-NKCC1 (T4, anti-NKCC1 cotransporter from human colonic crypt, T84 cell, developed by Lytle et al. 1995 [34], produced in mouse and obtained from the Development Studies Hybridoma Bank of the Department of Biological Sciences of the University of Iowa, USA) and the secondary antibody anti-mouse IgG (Fab specific) −FITC antibody, produced in mouse (Sigma Aldrich, Burlington, MA, USA). The T4 antibody has been shown to be specifically immunoreactive with NKCC from many vertebrates and invertebrates, (e.g., [35,36]). Negative controls for each species, incubating sections only with the secondary antibody, were carried out to confirm the specificity of the fluorescent signal and the binding of the primary antibody with the NKCC cotransporter along the body of annelids (Appendix A). Slides were mounted with Fluoromount-G Mounting Medium (Invitrogen) and conventional glass coverslips. To avoid fading, the slides were protected from light until the moment of observation with confocal microscopy. Slides were observed and photographed under a Nikon A1R MP+ multiphoton confocal microscope. Images from negative control sections were taken with the same settings (i.e., fluorescence intensity and exposure times) used for the positive ones. Due to the lack of further investigation focused on the functional characterization of the proteins in our target species, we considered the NKCC signal recovered in our results as an NKCC-like protein.

### 2.4. Quantification and Statistics

Depending on the species, the fluorescence intensity of a selected area of each picture from different parts of the body was quantified by using Fiji 1.0 software [37] on 7–10 specimens per species, including the negative controls. To test where the fluorescence was higher (i.e., higher abundance of an NKCC-like protein) and if it was significantly different from the negative control, we took the five sections (i.e., focal planes of the stacked image acquired with the confocal microscope) with highest gray values/pixel^2^ per area selected and used them as replicates (i.e., five pseudoreplicates per target tissue). The full dataset including the raw data is available in Appendix A. The use of five pseudoreplicates was preferred to the section with the highest gray values/pixel^2^ because it allowed us to assess the variability in each sample even when the replication in each microscopy session was low. The nonparametric Kruskal–Wallis test [38] was used to test for significant differences, in each species and in the same confocal session, between the negative control and the tissues (e.g., integument, epithelium, cirri, branchiae) treated with the primary antibody and among different body parts (e.g., head, body, parapodium, thorax). Afterwards, the non-parametric Wilcoxon test [39] was applied as a post-hoc analysis to reveal which parts were significantly more fluorescent. The analyses were run using the *kruskal.test* and *pairwise.wilcox.test* in R version 4.3.2 [40]. Due to the scattered data about the anatomy of the studied annelid species, histological slides of sagittal sections of specimens of each species were also taken, staining with periodic acid Shiff (PAS) and Mayer’s hemalum solutions following the protocol [41], to compare them with the immunofluorescence pictures and allow for a more accurate interpretation of the results.

## 3. Results and Discussion

The blastp search retrieved sequences of two annelids: *Capitella teleta* Blake, Grassle and Eckelbarger 2009, a widespread annelid usually occurring in intertidal and shallow-water habitats [42] and *Helobdella robusta* Shankland, Bissen and Weisblat 1992, a freshwater leech [43]. Both species belong to the Sedentaria clade [3]. The phylogeny reflected the actual reconstruction of metazoans (Appendix A), positioning the annelids together with a mollusc, i.e., *Lottia gigantea* Sowerby 1834 [44].

With the immunofluorescence, the signal of the NKCC-like cotransporter was revealed along the body of the two nereidids, the nephtyid (Errantia), and the melinnid (Sedentaria). Each species showed a different pattern of immunolocalization and expression of the cotransporter (Figure 3 and Figure 4), suggestive of physiological adaptations to their specific estuarine habitats and salinity regimes. In both *A. yarae* and *L. acuta*, a significant signal (*p*-value < 0.001) was retrieved in either the muscles, integument, or internal epithelia (Figure 3A,B,D), and in *A. yarae*, also in cephalic appendages (Figure 3C). In *A. yarae*, the integument and the muscles of the parapodia displayed a significantly higher signal (*p*-value < 0.05) when compared with the segments (Appendix A, Appendix A). Parapodia of some annelids (e.g., nereidids, capitellids) are involved in ionic, gas, and dissolved organic matter exchanges [2,24,45,46], suggesting a potential role in salt and water fluxes of this species parapodia. Moreover, parapodia are highly vascularized, bearing also cirri known to have sensory functions, together with cephalic appendages [47]. The involvement of the NKCC1 was already reported in sensory organs of some invertebrate taxa, such as molluscs, spiders, and flies [48,49,50], by modulating the effect of GABAergic transmissions with the transport of Cl⁻, important for synaptic transmissions [51].

*Laeonereis acuta*, the euryhaline species occurring in the poly- and euhaline sectors of the estuary, showed a higher signal of the NKCC-like protein along the integument than in its muscle bundles (*p*-value < 0.01; Appendix A, Figure 3F–H). The higher abundance of NKCC-like protein along the integument may be due to its position directly in contact with the external medium and, consequently, facing salinity fluctuations and volume changes. In fact, it is obvious that the main function of the cuticle is protective, being influenced in composition and complexity by the environment and the animal lifestyle [52]. *Laeonereis acuta* also showed a slightly higher signal in the parapodium integument than in the body, however, in one of the sessions this was not significant (*p*-value < 0.06); Appendix A). The distinct location of the NKCC-like protein along these two nereidids may be related to their varying life strategies and reflect their tolerance and responses to salinity variations, since *A. yarae* is usually found in hard substrata, while *L. acuta* is a tube-dweller of sandy/muddy bottoms. Previous studies on the effects of osmotic stress in these two species pointed out a stronger cell volume regulation and higher euryhalinity of *L. acuta* in comparison with other estuarine and marine annelids [4,5,53].

The NKCC-like signal in the nephtyid *Nephtys fluviatilis* was significantly higher in the integument, muscles, and peritoneum than in the negative controls (*p*-value < 0.001) but did not vary among tissues or body parts (Appendix A, Figure 4A–C). Considering the size of the parapodia, which increases from the head to the pygidium, the animals were roughly divided into anterior, central, and posterior parts. Interestingly, for the integument, the intensity of the fluorescence displayed by the posterior portion was statistically the highest among them (*p*-value < 0.01), while the central portion was the lowest (*p*-value < 0.01; Appendix A). Taking into account the life strategies of this species, such a pattern may reflect the varying exposure of each body part to varying osmotic stress. In fact, these burrowing animals perform regular vertical movements in the galleries they dig, depending on the tide [54]. Thus, they may leave their extremities more exposed to salinity fluctuations than the middle portion of their bodies. At the same time, a higher intensity in both the anterior and posterior parts may suggest other roles for this membrane protein in addition to osmoregulation. It may be potentially involved in processes regarding growth and regeneration, both related to the posterior part of the body [55,56]. Moreover, the anterior part of the body bears the brain as well as some sensory organs, such as the nuchal organs, that may require the presence of this protein for neurotransmission. In fact, besides neurotransmission and osmoregulation, a higher expression of NKCC1 was reported for neurogenesis of murines, body development (e.g., connective and olfactory tissues, early embryonal stages of developing heart), and brain injuries in mice [57,58,59].

In contrast with the other species, the intensity of the fluorescence in *I. pulchella* was significantly higher (*p*-value < 0.01) than that in the negative controls only in the branchiae and the internal tissues of either the thorax or the abdomen (Figure 4E–G). This may be the result of both its lifestyle and anatomy. *Isolda pulchella* is a tube-building suspension feeder that usually occurs among the roots and rhizomes of salt marshes in the polyhaline and euhaline sectors of estuaries. Like other terebellomorphs, the head and the branchiae—located in the first segments of the thorax—are more exposed to salinity fluctuations than other body parts because they are usually placed outside the tube to catch food [60]. The production of mucus may act in this case in a contrasting way. From a physiological point of view, *I. pulchella* may rely on mucus production to buffer the effects of salinity fluctuations along the length of its body, which would explain the absence of the NKCC-like protein in the integument. The mucous secretions were already found as an osmoregulatory response in other animals, such as oligochaetes and gastropods [61,62]. On the other hand, its production and presence along the body may have led to a bias in this work. Natural fluorescence was reported for the mucus secreted by some other annelid species [62,63], and the potential autofluorescence of the mucus of *I. pulchella* may have hindered the signal induced by the presence of the NKCC-like protein. Supporting this hypothesis, the intensity of the fluorescence of the negative controls of the integument was significantly higher than that of the treatments (Appendix A). Assuming the occurrence of autofluorescence, these results may suggest that the intensity of the NKCC-like signal was either lower than the mucus signal or completely absent and that the treatments interfered with the autofluorescence itself. However, potential autofluorescence may also be related to the cuticle itself, which is composed of collagenous fibers [52], known for the typical autofluorescence seen mostly throughout the green wavelengths [64]. The highest fluorescence signal was recovered in the inner tissues of the thorax (Appendix A, Figure 4F and Appendix A), which may reflect the position of the excretory system in this family. As with other species belonging to sedentary families (e.g., Sabellidae, Terebellidae), few pairs of protonephromixia are restricted to the anterior part of the body [7]. In addition, the first nephridiopore in *Isolda* is located in the IV segment between the branchiae [65,66], highlighting why the signal of the immunofluorescence may have been higher in the first segments of the thorax. Indeed, NKCC is known to be associated with the excretion system of vertebrates in apical membrane of kidney epithelial cells from the thick ascending limb of Henle’s loop (NKCC2) [16,17] as well as in gills of marine teleosts, rectal glands of elasmobranchs, and salt glands of marine birds and reptiles (NKCC1) [12,14,15,16,17]. Finally, the NKCC-like protein was retrieved as well in the digestive epithelia located in the abdomen (*p*-value < 0.001; Appendix A), supporting once again the similarity of functions (i.e., uptake of Cl^−^ and position of this membrane protein in other taxa, either vertebrates or invertebrates [20,35,66].

## 4. Conclusions

The NKCC-like cotransporter, retrieved in silico in two species of annelids, has been immunocytochemically recovered and localized in the bodies of four other species of estuarine annelids, balancing the number of confirmed annelids with this cotransporter between Sedentaria and Errantia (i.e., three species for each clade). Its expression was not homogeneous among the species examined, putatively reflecting their specific ecological challenges with respect to cell volume regulation. The free-living/burrowers (both nereidids and the nephtyid) displayed a more widespread signal for NKCC-like proteins, in contrast to the stenohaline and sedentary melinnid, which displayed a more limited signal to a few regions of the body. Moreover, the variety of tissues in which the signal was retrieved suggests its centrality to the crucial process of cell volume regulation in essentially conforming invertebrates that dwell in waters of potentially fluctuating salinities, stressing the need for further physiological studies to fully understand the role played by this membrane protein in annelids.

## Figures and Tables

**Figure 1 biology-13-00235-f001:**
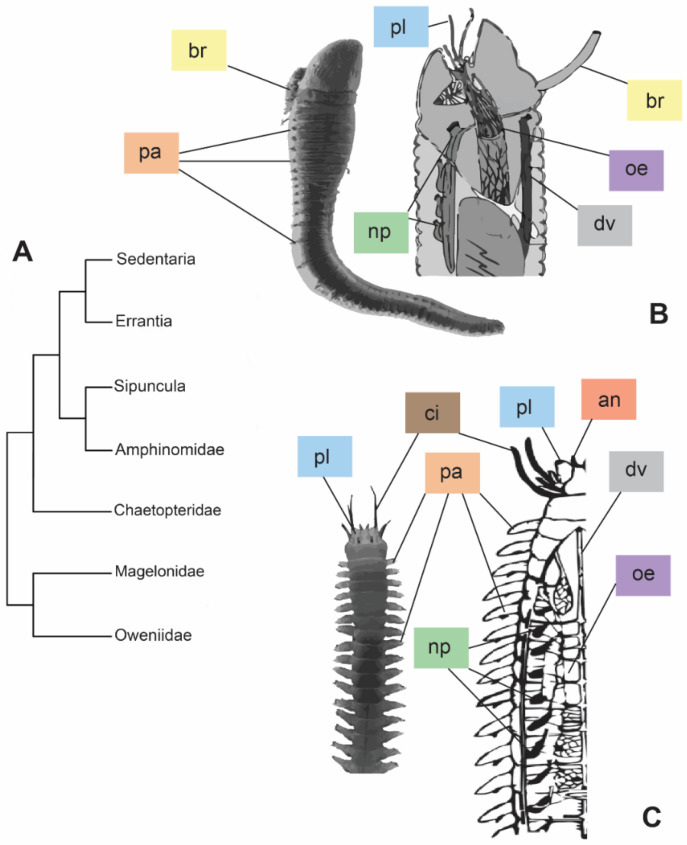
Phylogeny modified from [3] based on the molecular approach (**A**) and comparison between morphologies of the two major clades, Sedentaria (**B**) and Errantia (**C**). (**B**) = entire individual of *Artacama* sp. (Terebellidae) and sagittal section of an Alvinellidae anterior part. Images modified from [8,9], respectively. (**C**) = entire individual of *Neanthes* sp. (Nereididae) and frontal section of a Nereididae. Images modified from [10,11], respectively. Abbreviations: an = antennae; br = branchiae; ci = cirri; dv = dorsal vessel; oe = oesophagus; np = nephridia; pa = parapodia; pl = palps. Colors of the labels referring to the different organs.

**Figure 2 biology-13-00235-f002:**
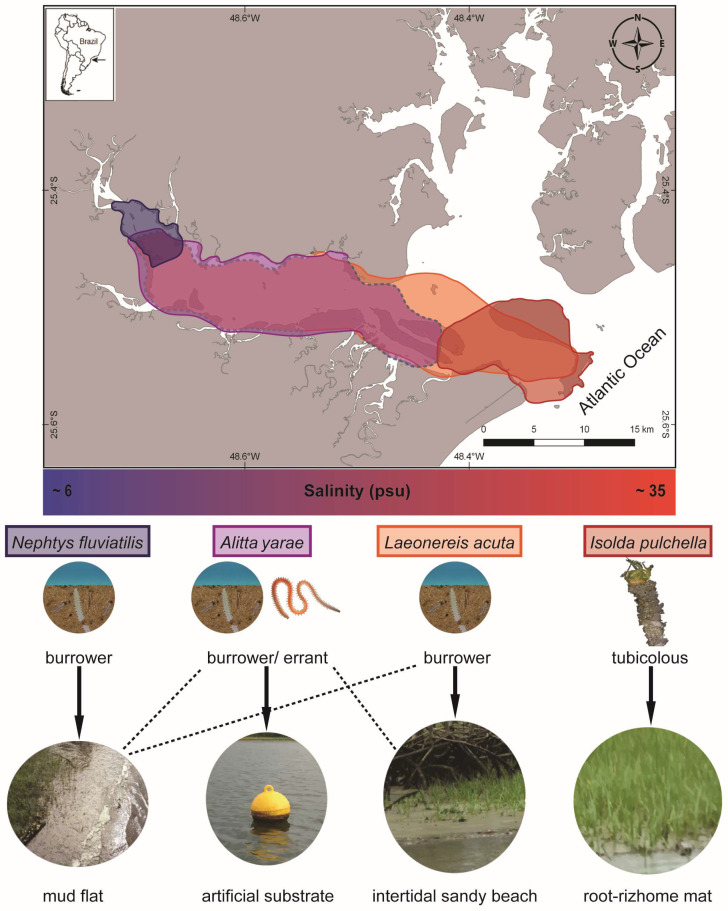
Map representing the occurrence of the studied species and a simplified gradient of salinity along Antonina and Paranaguá Bays, life strategies, and types of habitats. Colors on the map referring to the area of occurrence: blue *N. fluviatilis*, violet *A. yarae*, orange *L. acuta*, red *I. pulchella*. Dashed lines in the map and in the types of habitats indicate occasional occurrences.

**Figure 3 biology-13-00235-f003:**
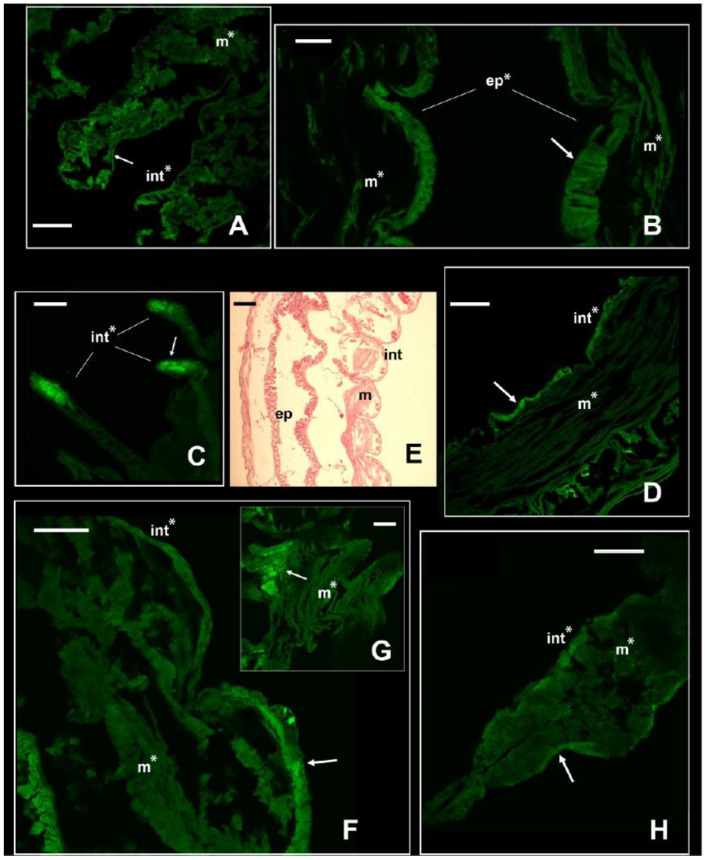
Signal of NKCC in the two euryhaline annelids *Alitta yarae* and *L. acuta*. Immunofluorescence signal in *Alitta yarae* is shown in (**A**–**D**); in *L. acuta*, it is shown in (**F**–**H**). Arrows pointing to the most prominent staining. Significant comparisons with the negative control represented with an asterisk (*: *p* < 0.05). Abbreviations: ep = internal epithelium; int= integument; m = muscles. (**E**) = Sagittal histological section of a specimen of *L. acuta* for comparison with immunofluorescence pictures. *Alitta yarae*: (**A**) = parapodia; (**B**) = digestive trait; (**C**) = tentacular cirri; (**D**) = segments. *L. acuta*: (**F**) = segment; (**G**) = detail of the muscles; (**H**) = parapodium. Scale bars ((**A**–**C**) and (**D**–**H**)): 160 µm; scale bar (**E**): 500 µm.

**Figure 4 biology-13-00235-f004:**
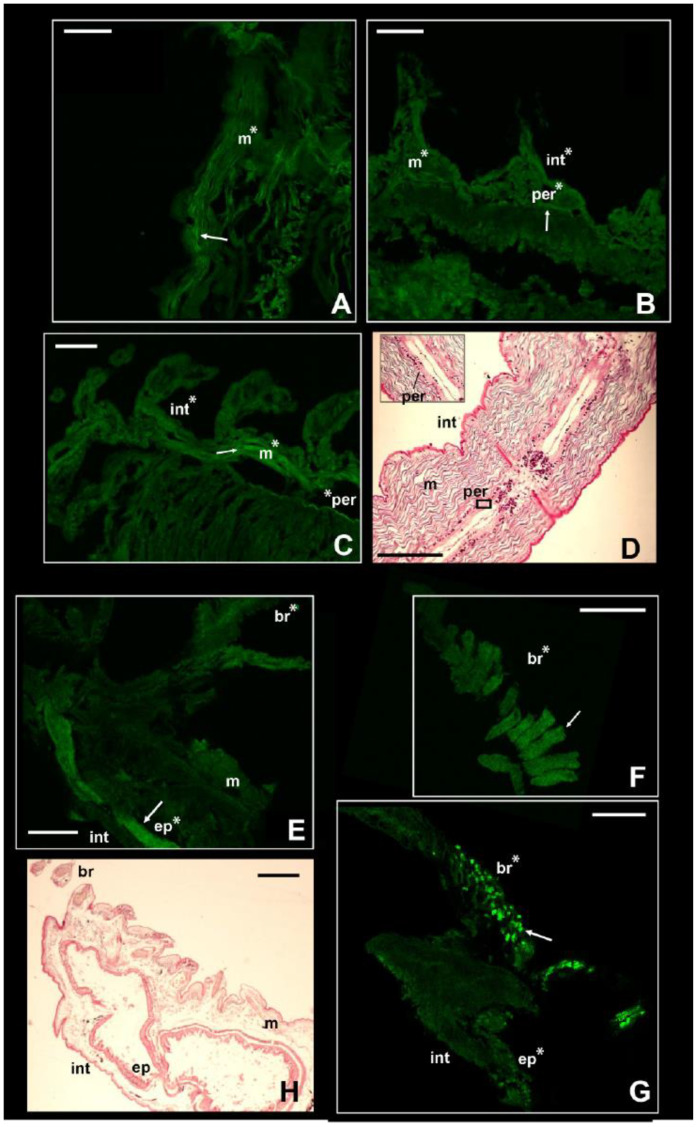
Signal of NKCC in the two stenohaline annelids *N. fluviatilis* and *I. pulchella*. Immunofluorescence signal in *N. fluviatilis* shown in (**A**–**C**); in *I. pulchella* in (**E**–**G**). Arrows pointing to the most prominent staining. Significant comparisons with the negative control represented with an asterisk (*: *p* < 0.05). Abbreviations: br = branchiae; ep = internal epithelium; int = integument; m = muscles; per = peritoneum. (**D**,**H**) = Sagittal histological sections of a specimen of *N. fluviatilis* and *I. pulchella*, respectively, for comparison with immunofluorescence pictures. *N. fluviatilis*: (**A**) = detail of a segment; (**B**) = posterior segments with parapodia; (**C**) = central segments with parapodia and peritoneum. *I. pulchella*: (**E**,**G**) = head and first segments of the thorax; (**F**) = detail of the branchia. Scale bars (**A**–**H**): 160 µm.

## Data Availability

The data presented in this study are available in Appendix A.

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
