# Peer review of "First Insights into Body Localization of an Osmoregulation-Related Cotransporter in Estuarine Annelids"

_biology, 2024, doi:10.3390/biology13040235_

Round 1
Reviewer 1 Report
Comments and Suggestions for Authors
Review for the paper “First insights into body localization of an osmoregulation-related cotransporter in estuarine annelids” by Serena Mucciolo, Andrea Desiderato, Maria Mastrodonato, Paulo Lana, Carolina Arruda Freire, Viviane Prodocimo submitted to “Biology”.
In a laboratory study, the authors tested the hypothesis of the presence of the Na⁺-K⁺-2Cl- cotransporter in annelid tissues to determine how various stimuli affect the behavior of shore crabs. The authors detected this cotransporter in silco and then detailed their results using immunofluorescence. They found the NKCC signal in different tissues of four estuarine annelid species from the Paranaguá Estuarine Complex, southern Brazil. The authors found different levels of expression of the NKCC-like protein and suggested that this result might be related to different salinity tolerance of the studied species, reflecting their preferred habitat conditions. These novel findings expand our knowledge of the body localization of NKCC in polychaetes and provide a basis for further research. The authors used standard sampling methods and laboratory procedures and equipment to accomplish the objectives of this study. The data were statistically processed and the results were discussed based on previous findings. Some revisions are needed to improve the paper.
The authors should include information about the study site in the title, simple summary, and abstract.
L 71. Change “(Preston, 2009)” to “[2]”.
L 115-117. This statement is not entirely correct. True marine annelids are adapted to ocean salinity and cannot be classified as euryhaline. In addition, the authors provide two inappropriate references: one based on a study of mollusks and the other on a study of crustaceans, not annelids. The authors should correct this text and provide more relevant citations.
In the Materials and methods section, the authors should specify the sampling period, including the year and month of sampling. They should also indicate the environmental conditions of the sampling period (dry or wet, colder or warmer, etc.).
Did the authors measure salinity at the time of sampling? What source was used to create the salinity distribution on the map?
The authors should report the sample sizes for their study: how many specimens of each species were collected and examined (including negative control)? Did the authors randomly subsample the worms? Did the authors measure the length and weight of the polychaetes? It would be useful to present these data in the paper.
It is very difficult to follow the text without the figures showing the main patterns of fluorescence in annelid tissues. I think the authors should include these figures in the main text instead of in the supplementary material. Also, the authors should use different superscript letters above the boxes to highlight significant differences between pairs of tissues.
The text should be supplemented with the corresponding H, W, and p values for statistical tests.
References should be updated with the required details, including volume number and pages (or article number) for journal publications.
The authors should also revise and italicize Latin names as appropriate.
Comments on the Quality of English LanguageMinor revision
Author Response
Reviewer 1
A: We thank the Reviewer for all the comments and suggestions. We accepted most of them and edited the text accordingly. Below, the detailed answers to each comment.
R: The authors should include information about the study site in the title, simple summary, and abstract.
A: We partially agreed with the Reviewer. We added this information in both the simple summary and the abstract (check lines 24 and 34, respectively), however, we think that the study area (southern Brazil) is not the focus of the manuscript, therefore, we decided to not add it in the title as the environment (i.e., estuarine) is more important.
R: L 71. Change “(Preston, 2009)” to “[2]”.
A: Thank you for pointing this out; we changed the reference accordingly.
R: L 115-117. This statement is not entirely correct. True marine annelids are adapted to ocean salinity and cannot be classified as euryhaline. In addition, the authors provide two inappropriate references: one based on a study of mollusks and the other on a study of crustaceans, not annelids. The authors should correct this text and provide more relevant citations.
A: We agreed with the Reviewer. We corrected the sentence by changing the term “marine annelids” with “estuarine annelids” (line 116). We also changed the citations, focusing only on annelids (see references n. 2, 22, 23, in lines 383, 421, 425, respectively).
R: In the Materials and methods section, the authors should specify the sampling period, including the year and month of sampling. They should also indicate the environmental conditions of the sampling period (dry or wet, colder or warmer, etc.).
A: The information was added in the text (line 148: “Sampling was carried out during the dry season (May-September 2018) along the Paranagua` Estuarine Complex (PEC) in southern Brazil, […]).
R: Did the authors measure salinity at the time of sampling? What source was used to create the salinity distribution on the map?
A: Yes, the salinity was measured at the time of sampling. The salinity distribution on the map is based on data about the average salinity recorded during other sampling along all the estuary, plus publicly available literature. However, we are aware that this map represents a simplification of the salinity fluctuations along the estuary, without taking into account daily and seasonal changes. Thus, we edited the figure caption, hoping to clarify (check lines 165-166: “Map representing the occurrence of the studied species and a simplified gradient of salinity along Antonina and Paranagua` Bays, life strategies and types of habitat”).
R: The authors should report the sample sizes for their study: how many specimens of each species were collected and examined (including negative control)? Did the authors randomly subsample the worms? Did the authors measure the length and weight of the polychaetes? It would be useful to present these data in the paper.
A: A total of 7-10 specimens per species was examined, including the negative controls (this information is now corrected in the text; check lines 204-205). However, during the sampling we collected more specimens (~20) to make sure to have a minimum number of replicates in case of death during the acclimatization. Unfortunately, we did not weigh the animals, but we examined only adult specimens with a similar size (we added in the text the average length of each species; check lines 176-177), that looked healthy and were not regenerating. Now, it is also specified in the text in line 173 (“Healthy adults were chosen as described in Mucciolo et al. 2021 [5]”), by citing the work where we firstly described in detail how we selected the animals for other experiments.
R: It is very difficult to follow the text without the figures showing the main patterns of fluorescence in annelid tissues. I think the authors should include these figures in the main text instead of in the supplementary material. Also, the authors should use different superscript letters above the boxes to highlight significant differences between pairs of tissues.
A: We appreciate the suggestion of the Reviewer; however we think that these pictures would not improve the readability of the manuscript. Nevertheless, we agree that the significance of the comparisons should be shown in the main text, and we included asterisks in the pictures of the fluorescence. Also, we improved the pictures in the supplementary material and included asterisks for the difference with the negative control and the letters or the pairwise significant comparisons.
R: The text should be supplemented with the corresponding H, W, and p values for statistical tests.
A: We agree and included the p-values in the text.
R: References should be updated with the required details, including volume number and pages (or article number) for journal publications.
Done.
R: The authors should also revise and italicize Latin names as appropriate.
A: Done.
Reviewer 2 Report
Comments and Suggestions for Authors
Well done. Please separate Results and Discussion. Novelty paper always fares better if this is done.

Minor syntax & wording problems.
Author Response
Reviewer 2
Well done. Please separate Results and Discussion. Novelty paper always fares better if this is done.
We thank the Reviewer for all the comments; we accepted most of the suggestions and edited the text accordingly. Below, the detailed answers to each comment.
R: L. 68 (now L. 69): life in tubes?
A: Yes, several families of annelids are sessile (e.g., serpulids, sabellids, terebellids) and spend their life in the tubes they build. Accordingly, the organization of the body may change (e.g., number and position of nephridia, position of the anus, presence of specialized glands). An example is provided by Tarallo et al. 2016, where the authors explained these differences in several annelid taxa, and both compared the metabolic rate and the DNA base composition of annelids with different lifestyles. Check also Merz 2015, for an extended comparison of polychaete tubes composition and structures, with a discussion on how it may be related to the morphology of the polychaete.
References:
Merz 2015: https://doi.org/10.1111/ivb.12079
Tarallo et al. 2016: https://doi.org/10.1152/physiolgenomics.00018.2016
R: L. 142 (now L. 144): Why this particular group and species were chosen? Convenience?
A: Our in silico search was based on the comprehensive analysis of the entire CCC family carried by Hartman et al 2013, where the sequences of the choanoflagellate Monosiga brevicollis KCC and CIP1 were used as outgroup. Similarly, we used these sequences (i.e., Monosiga brevicollis KCC and CIP1) as outgroup to efficiently root our phylogeny and included the putative NKCC annelid ones, to see if our results were matching with the ones obtained by Hartman et al 2013.
R: L. 165: Hard to believe that salinity can be presented as linear change (simply a function of longitude).
A: The map shows the average salinity gradient along the bays; however, we agree with the Reviewer that it is a simplification of it without considering the daily and seasonal salinity fluctuations. To leave it clearer, we slightly changed the figure caption (check lines 165-166: “Map representing the occurrence of the studied species and a simplified gradient of salinity along Antonina and Paranagua` Bays, life strategies and types of habitat”). Nevertheless, indeed the Paranagua` Estuarine Complex as an almost longitudinal gradient of salinity given its extension, and this has already been reported in several papers (e.g., Lana et al. 2001, Marone et al. 2005).
References:
Marone et al. 2005: https://doi.org/10.1590/S1679-87592005000200007
Lana et al. 2001: https://doi.org/10.1007/978-3-662-04482-7_11
R: L. 215 (now L. 219): This needs better explanation: how exactly your interpretation will be enhanced by this additional information?
A: With immunofluorescence, the only parts of the body that are visible at the confocal are those were the antibody signal is recovered (i.e., fluorescent). Therefore, not all the tissues may be visible, leading to some problems with the correct interpretation of the observed part of the body. However, we reformulated the sentence hoping to leave it clearer (lines 219-225, now: Due to the scattered data about the anatomy of the studied annelid species, histological slides of sagittal sections of specimens of each species were also taken staining with periodic acid Shiff (PAS) and Mayer’s hemalum solutions following the protocol [41], to compare them with the immunofluorescence pictures and allow for a more accurate interpretation of the results.”).
R: L. 218 (L. 226): I strongly recommend that you separate your Results and your Discussion.
A: We appreciated the suggestion of the Reviewer to separate the results and discussions, but being a short communication, we decided to keep them combined to discuss each result we obtained and leave the text easier to follow.
R: L. 232 (L. 237): significance stats?
A: Included.
R: L. 263 (L. 273): significance stats?
A: Included.
R: L. 284 (L. 294): significance stats?
A: Included.